# Perceive before Respond: Improving Sticker Response Selection by Emotion Distillation and Hard Mining

Submission Id: 2084

## ABSTRACT

In online chatting, people increasingly prefer using stickers to supplement or replace text for replies, as sticker images can express more vivid and varied emotions. The Sticker Response Selection (SRS) task aims to predict the sticker image that is most relevant to the history dialogue. Previous researches explore the semantic similarity between context and stickers, while ignoring the role of both unimodal and cross-modal emotional information. In this paper, we propose a **"Perceive before Respond"** (PBR) training paradigm. PBR perceives sticker emotions through a knowledge distillation module. Variety sticker representations of each emotion category are acquired from the large-scale sticker emotion recognition dataset and distilled into our model to enhance emotion comprehension. To make a better response, we further distinguish stickers with similar subject elements under the same topic. We perform contrastive learning at both inter-topic and intra-topic levels to obtain discriminative and diverse sticker representations. In addition, we improve the hard negative sampling method for image-text matching based on cross-modal sentiment association, conducting hard sample mining from both semantic similarity and sentiment polarity similarity for sticker-to-dialogue and dialogue-to-sticker. Extensive experiments verify the effectiveness of each proposed component. Ablation experiments on different backbone networks demonstrate the generality of our approach. **The code is provided in the supplement material and will be released to the public.**

## CCS CONCEPTS

• **Computing methodologies → Artificial intelligence**; • **Information systems → Sentiment analysis**.

## KEYWORDS

sticker response selection, multimodal learning

## 1 INTRODUCTION

With the rapid development of instant messaging applications, stickers are widely used in online chats. Compared to simple emoticons (*e.g.*, emojis), stickers are more expressive and superior in conveying strong emotion, positivity, and intimacy [22]. It plays an important role in assisting people to express emotions and regulating the emotional tendencies of the conversation [5, 45, 57]. As shown in Fig. 1 (a), the Sticker Response Selection (SRS) task uses the historical context of multi-turn dialogue to recommend the sticker. With the popularity of sticker usage, SRS receives widespread attention from researchers [4, 22, 30, 33, 40, 45, 55]. It contributes to users engaging in vivid, expressive online chat and holds promising prospects for developing anthropomorphic intelligent robots [1, 16].

Compared to general text-image retrieval tasks, the dialogue and stickers in SRS exhibit stronger emotional relevance [22]. We extract the sentiment polarity scores of stickers and dialogues separately to

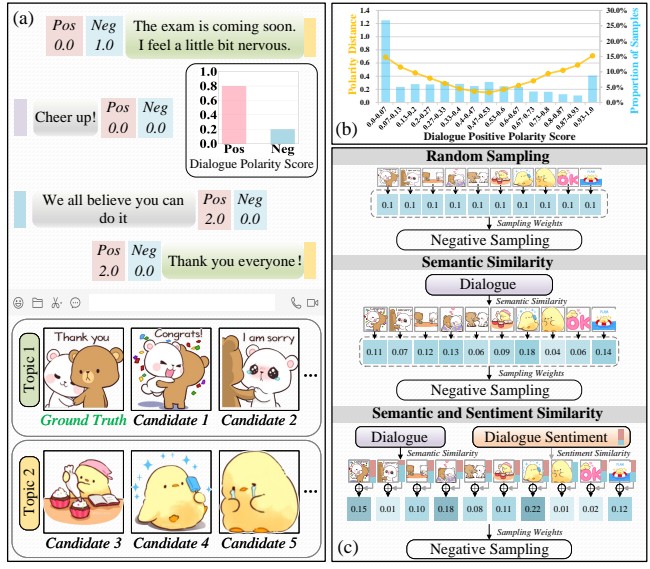

Figure 1: (a) Visualization example of the SRS task. The sentiment of the sticker is closely related to the dialogue history. (b) Dialogue-sticker sentiment verification. The blue bars indicate the number of samples, and the black line indicates the average Euclidean distance between dialogue and sticker sentiment scores. The horizontal axis represents the emotional scores of the dialogue, where 0 indicates the lowest positivity and 1 indicates the highest positivity. (c) Sampling probability with different hard sample mining strategies.

verify the cross-modal sentiment association. As shown in Fig. 1 (b), we divide the samples into 15 intervals based on their dialogue sentiment and count the number of samples in each interval. The orange lines indicate the average Euclidean distance between dialogue and sticker sentiment polarity. The upper limit of the distance is $\sqrt{2}$ and the average distance of all intervals is below 0.8, which indicates the sentiments of sticker and dialog are consistent in most samples.

To better model and leverage emotional information, three challenges need to be addressed in SRS. (1) The varied visual concepts across different topics present a challenge for understanding emotions. As shown in Fig. 2 (a), stickers with different semantic content may express the same emotion, while those with similar semantic content may convey different emotions. This is known as the Affective Gap for visual emotion analysis [56]. Additionally, existing SRS datasets lack emotion annotations, which exacerbates the difficulty of extracting emotional information from stickers. (2) Modeling discriminative representations for stickers is challenging. Due to the distinctive organization structure of stickers [32], images under the same topic are usually composed of the same subject elements and

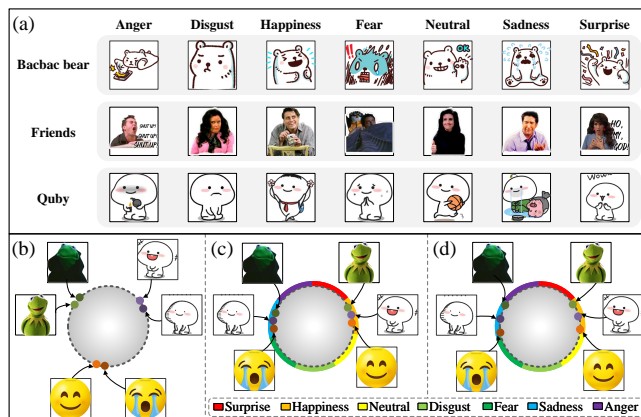

**Figure 2: (a) The Affective Gap exists in stickers. (b) The sticker representations obtained by previous methods are aggregated based on their topic. (c) EKD separates sticker features based on emotion categories. (d) TSCL further distinguishes features at the topic level.**

the extracted features are similar (*e.g.*, Fig. 2 (b)). This can hinder the SRS model from selecting appropriate stickers for the history dialogue, especially if the candidate stickers all belong to the same topic. (3) Most related works [12, 55] consider the SRS as a text-image matching problem, where the quality of negative sampling plays a crucial role in the performance of the multimodal model. Taking the negative sampling of dialogue-to-sticker as an example, as shown in Fig. 1 (c), the random selection approach leads to a uniform distribution of easy negatives and hard negatives. A widely used improvement approach is to select samples with high semantic similarity as hard negative samples [3, 24, 25]. However, in the SRS task, the difficulty of sample discrimination is also closely related to emotion. To further improve the quality of negative sampling, emotions are valuable which has been ignored in previous works.

In this work, we propose a *"Perceive before Respond"* (PBR) training paradigm. To identify the emotions conveyed by stickers, we introduce an Emotion Knowledge Distillation (EKD) module, which utilizes emotion knowledge obtained from the large-scale sticker emotion recognition dataset SER30K [32]. Inspired by the ClusterFit algorithm [52], we cluster the image features in SER30K, and call the centroid vector of each cluster Emotion Anchor. To describe images with diverse styles and semantic content in each emotion category, we set $M$ Emotion Anchor for each emotion category. For a sticker image, we compute its similarity to the Emotion Anchors to obtain emotion pseudo-labels. The image encoder learns emotional information guided by these pseudo-labels (*e.g.*, Fig 2 (c)). To better model the discriminative representations of stickers, we design a Topic-level Semantic Contrastive Learning (TSCL) module consisting of intra-topic and inter-topic two parts. For intra-topic contrastive learning, negative samples are selected within the same topic, aiming for the model to focus on the different local emotional features among stickers with similar subject elements. For inter-topic contrastive learning, negative samples are chosen from different topics, intending for the model to learn diverse representations of stickers across various topics (*e.g.*, Fig 2 (d)). Based on the consistency

between the semantics and emotions of stickers and dialogue, we designed a Polarity-based Hard Sample Mining (PHSM) module. For hard sample mining in the text-image matching problem, we consider cross-modal sentiment polarity similarity in addition to semantic similarity. Samples with higher similarity in both semantic and sentiment are considered hard samples for the model to discriminate. Assigning higher weights to such samples enables the SRS to mine more informative negatives.

The main contributions of this work are three-fold: (1) We propose the EKD module and TSCL module to exploit the emotional and semantic content of stickers and improve the quality of sticker features. (2) To the best of our knowledge, this is the first time that both dialogue and sticker emotions have been utilized in the SRS task. We design a PHSM module to mine hard samples by integrating the semantic and sentiment similarity between modalities. (3) The experimental results show that our method boosts the performance over the previous state-of-the-art methods by a large margin. Extensive ablations also verify the effectiveness of each designed module.

## 2 RELATED WORK

***Sticker Response Selection.*** With the development of online chat, the SRS task getting increasing attention in recent years [11–13, 23, 48, 55]. Laddha *et al.* [23] propose a sticker recommendation method that first predicts the next message, then replaces the predicted text with a sticker. Gao *et al.* [12, 13] propose a large-scale real-world dialogue dataset with stickers. The deep interaction network is used to match stickers and dialogue history, while the fusion network is used to fuse features and output the match score. As the extension work of [12], [13] considers the user's sticker preferences by additionally recording the user's historical dialogue information with a key-value memory network. Fei *et al.* [11] model sub-tasks such as text generation and sticker prediction as general sequence generation. A unified generation network is then applied to retrieval stickers. Wang *et al.* [48] design a multimodal encoder for dynamic GIFs, training in a similar way to CLIP [41].

***Multi-turn Dialogue Emotion Analysis.*** To recognize the sentiment of dialogue, some studies model the contextual dependencies between utterances. Li *et al.* [28] propose a bidirectional sentiment recurrent framework for contextual modeling and sentiment classification via a two-channel classifier. Yang *et al.* [53] introduce curriculum learning to train conversations from easy to hard. Ma *et al.* [34] detect the sentiment of dialogues from word and utterance level views. Some other studies focus on modeling interactions between speakers. Majumder *et al.* [35] model the speakers individually and use a recurrent neural network to track the state of each individual. Ghosal *et al.* [14] present the Dialogue Graph Convolutional Network to model dialogue context by exploiting both self-speaker and cross-speaker dependency. Ishiwatari *et al.* [21] add position encoding to relational graph attention networks, which capture both the dependencies between speakers and the sequential relationships between utterances.

***Knowledge Distillation.*** Knowledge distillation enables the small student model to learn knowledge from the large teacher model [19]. The offline distillation with the flexibility to choose pre-trained teacher models is more concerned with knowledge transfer. For

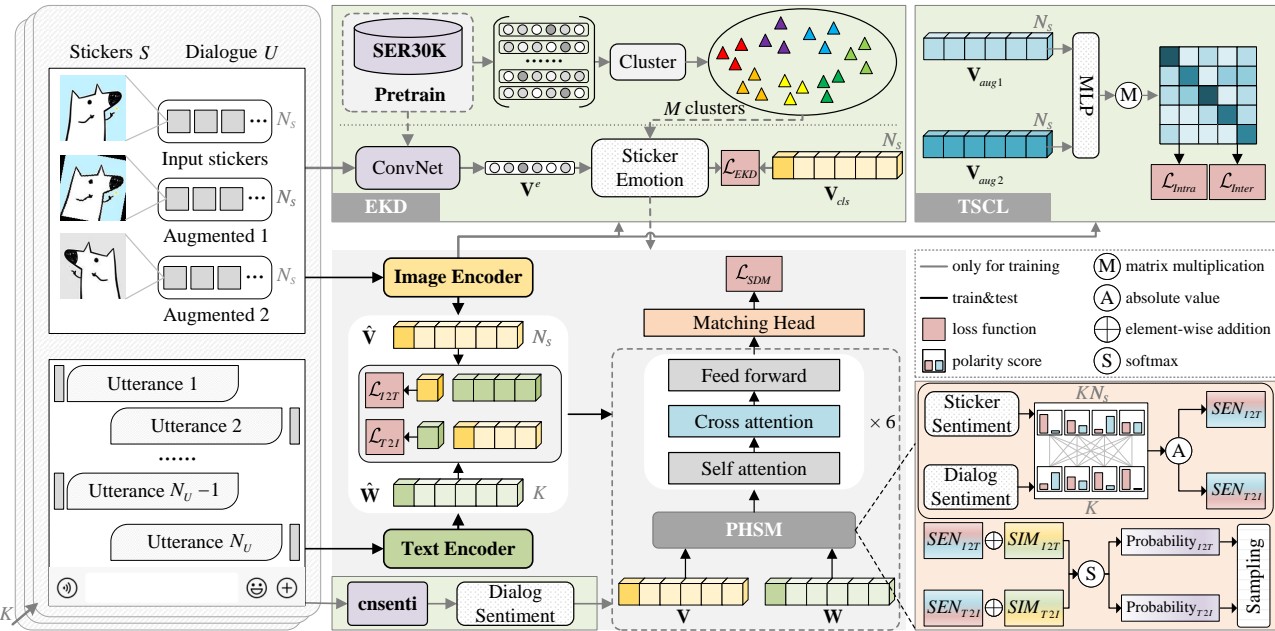

**Figure 3: Overview of our proposed approach. The overall model consists of an image encoder, a text encoder, and a multimodal encoder. The input sticker and dialogue are first sent to the unimodal encoders for feature extraction. We design Emotion Knowledge Distillation (EKD) and Topic-level Semantic Contrastive Learning (TSCL) modules, which improve sticker representations in terms of feature diversity and discriminative aspects, respectively. The unimodal features are then fed to the multimodal encoder after cross-modal alignment. We model the ranking of candidate stickers as a dialogue-sticker matching task. The Polarity-based Hard Sample Mining (PHSM) module mines hard samples that are similar in both semantics and sentiment and treats them as negative samples.**

knowledge design, Hinton *et al.* [19] use soft labels predicted by teacher models as category probabilities to train student models. Romero *et al.* [43] use the middle layer features of the teacher model as hints to distill a deeper and narrower student model. Zhang *et al.* [54] propose wavelet knowledge distillation to extract only the high-frequency band of images after discrete wavelet transformation. In terms of loss function design for feature alignment, Li *et al.* [27] align the output of $1 \times 1$ convolutional layers at the end of each block of the student network and the output of the teacher network to achieve fast fine-tuning. Mirzadeh *et al.* [36] add Teacher Assistant to mitigate the gaps between the teacher model and the student model.

***Hard Sample Mining.*** Informative hard samples can improve the model's performance [6]. Sample selection and sample reweighting are commonly used methods. Robinson *et al.* [42] prefer to select informative negative pairs with similar representations in contrastive learning. Lin *et al.* [31] reduce the weight of simple samples to solve the foreground and background imbalance problem in object detection. Different tasks have different measurements of sample difficulty. For example, long sentences are usually considered hard samples in NLP [39], while images containing more objects are considered hard samples in semantic segmentation [49]. In image-text retrieval models [25], hard samples are typically defined as samples with high contrastive similarity. For the SRS task, we introduce the

sentiment polarity of dialogue and sticker, measuring the sample difficulty from both sentiment and semantic views.

## 3 METHODOLOGY

### 3.1 Framework Overview

As shown in Fig. 3, our method mainly consists of an image encoder, a text encoder, and a multimodal encoder. We will first explain the problem definition of SRS and then introduce the details of each component.

**Problem Definition.** Given a dialogue history consisting of utterances from multiple users and a set of candidate sticker images, the Sticker Response Selection task aims to comprehend both the semantic and sentiment information in the dialogue history and select the best matching sticker. Formally, for the given dialogue $U = \{u_1, ..., u_{N_U}\}$ with $N_U$ utterances and $N_S$ candidate sticker images $S = \{s_1, ..., s_{N_S}\}$, our goal is to train a ranking model $f$ that assigns the highest score to the ground truth sticker:

$$pos = \arg\max_i f\left(s_i \mid U, s_i\right), s_i \in S \qquad (1)$$

Note that each candidate set has only one ground truth sticker, and assume that its index is $pos$.

**Image Encoder.** For a fair comparison, we employ three backbone networks as image encoders, *i.e.*, Inceptionv3 [44], ResNet [18], and ViT [10]. For each input $U$ and $S$ mentioned above, where $s_i \in \mathbb{R}^{H \times W \times 3}$ is an RGB image. For Inceptionv3 and ResNet, we

take the feature map after global pooling as the image representation and flatten it to a one-dimensional sequence. For ViT, we take the output of the last transformer encoder layer as the image representation. We define the sequence feature obtained from the encoder as $\mathbf{V} = \{v_{cls}, v_1, ..., v_{N_V}\}, v_i \in \mathbb{R}^{C_V}$, where $v_{cls}$ is the extra [CLS] embedding (*i.e.*, the global image feature), $N_V$ is the length of sequence features, $C_V$ is the number of dimensions of the feature. Specifically, for Inceptionv3 and ResNet, we take the global average of the last feature map as $v_{cls}$.

**Text Encoder.** We also use four different backbone networks to obtain the dialogue representation, *i.e.*, LSTM [20], GRU [7], Transformer [47], and BERT [9]. For the dialogue history, we combine each utterance and use the [SEP] token to indicate their separation. Then we extract the text representations $\mathbf{W} = \{w_{cls}, w_1, ...w_{N_T}\}, w_i \in \mathbb{R}^{C_T}$, where $N_T$ is the number of words, $C_T$ is the dimension of word embedding. Specifically, for LSTM and GRU, we also use the average of all word features as $w_{cls}$.

To obtain compact multi-modal features, following [25], we employ contrastive learning to align visual and textual representations before fusing them. We first map the features of vision and language modalities to a common embedding space:

$$\hat{\mathbf{v}} = \mathbf{V}W^I, \hat{\mathbf{w}} = \mathbf{W}W^T, \qquad (2)$$

where $W^I \in \mathbb{R}^{C_V \times C}$, $W^T \in \mathbb{R}^{C_T \times C}$.

For image-to-text alignment, suppose a mini-batch has $K$ samples, and for the $s_{pos}$ in the $i$-th sample we combine it with $U_i$ to form a positive pair. The remaining $K-1$ dialogues are combined with $s_{pos}$ to form the negative pairs. We use the InfoNCE loss [37] to optimize our model, which is defined as

$$\mathcal{L}_{I2T} = \frac{-1}{K} \sum_{i=1}^{K} \log \frac{\exp\left(\hat{v}_{pos}^i \cdot \hat{w}^i\right)}{\sum_{j=0}^{K} \exp\left(\hat{v}_{pos}^i \cdot \hat{w}^j\right)}. \qquad (3)$$

For text-to-image alignment, the negative samples of text $U_i$ are not other stickers from the same batch, but negative sample stickers from the candidate set $S_i$:

$$\mathcal{L}_{T2I} = \frac{-1}{K} \sum_{i=1}^{K} \log \frac{\exp\left(\hat{w}^i \cdot \hat{v}_{pos}^i\right)}{\sum_{j=0}^{N_S} \exp\left(\hat{w}^i \cdot \hat{v}^j\right)}, \qquad (4)$$

where both $\hat{v}$ and $\hat{w}$ are [CLS] tokens in Equ. (3-4), and we omitted the subscript *cls* for simplicity.

**Multimodal Encoder.** The multimodal encoder uses the last six layers of the BERT. The difference between it and the text encoder is the application of the Cross Attention (CA) operation in each layer. The CA is similar to Multi-head self-attention, but the Key and Value embeddings are from the image, and Query embedding is from the text. Finally, we use the [CLS] token output from the multimodal encoder to predict the matching score between each sticker and dialogue. We model that as a binary classification task and optimize it using cross-entropy loss:

$$\mathcal{L}_{SDM} = \frac{-1}{K} \sum_{i=1}^{K} \left(\log \mathbf{P}\left(S_{pos}, U_i\right) + \log\left(1 - \mathbf{P}\left(S_{neg}, U_i\right)\right)\right), \qquad (5)$$

where $\mathbf{P}$ is the predicted probability, *neg* denotes the negative samples, and we will elaborate on the negative sample selection strategy in the subsequent sections.

## 3.2 Emotion Knowledge Distillation

Sticker images have potential emotion information. We extract sticker emotion knowledge from a recently proposed sticker emotion recognition dataset SER30K [32] and transfer them to our image encoder. Formally, we chose ResNet [18] as the teacher model and trained it on SER30K. The final sentiment classification accuracy of the teacher model on the SER30K test set is 64.56%, and the details of the teacher model are shown in the supplementary material. Then we first use the teacher model to extract a $C_S$-dimensional feature representation for each image in SER30K and group them according to seven emotion categories (*i.e.*, surprise, happiness, disgust, fear, sadness, anger, neutral). Images with different topics appear in the same emotion category. These images share the same emotion but can vary significantly in content and style. Using a single feature cannot accurately describe one emotion category. To address this issue, we employ the K-Means [17] to cluster image features of each emotion class into M clusters. The centroid vectors of each cluster are aggregated together to obtain a feature matrix $E \in \mathbb{R}^{C_S \times 7M}$, which we call the Emotion Anchor. Utilizing Emotion Anchor we can comprehensively characterize the images in each emotion category.

In the training phase, we first extract a feature representation $\mathbf{V}^e$ for each new sticker image using the teacher model mentioned above. Then its dot product result with Emotion Anchor is processed by softmax as the emotion pseudo label. To achieve emotion knowledge distillation, we minimize the KL divergence between the image encoder output and the emotion pseudo label:

$$\mathcal{L}_{EKD} = \mathrm{D}_{KL}(\mathbf{V}W^E, \mathrm{Softmax}(\mathbf{V}^e E)), \qquad (6)$$

where $W^E \in \mathbb{R}^{C_V \times 7M}$ is a linear transformation.

## 3.3 Topic-level Semantic Contrastive Learning

Stickers under the same topic have similar subject elements, which hinders the model from learning discriminative features. To address the challenge, we propose an intra-topic semantic contrastive learning to pull apart the image representations under the same topic. On the other hand, image representations under different topics should be naturally diverse, thus we also designed the inter-topic semantic contrastive learning branch to enhance such diversity.

*1) Intra-Topic Semantic Contrastive Learning.* As mentioned in Fig. 2, sticker images under the same topic have similar subject elements, and traditional contrastive learning based on semantic labels hinders the model from learning discriminative features. However, stickers in the same topic usually express diverse emotions. we perform contrastive learning within the topic to pull apart image representations while improving the ability of the model to capture local emotional features.

For the sticker images under the same topic, we perform two parallel data augmentations and get the output of the image encoder as $\mathbf{V}_{aug1}$ and $\mathbf{V}_{aug2}$. We first project the [CLS] token features $v_{aug1}, v_{aug2}$ of each sticker to a semantic representation space using a nonlinear transformation $\psi$ (*i.e.*, MLP with a ReLU function). Then we utilize the images under the same topic as negative samples and define the learning objective of intra-topic semantic contrastive

---

**Algorithm 1:** Polarity-based Hard Sample Mining

**input** : $SIM_{I2T} \in \mathbb{R}^{K \times K}$, $SIM_{T2I} \in \mathbb{R}^{K \times KN_S}$, $P^T \in \mathbb{R}^K$,
$\quad\quad\quad N^T \in \mathbb{R}^K$, $EMO \in \mathbb{R}^{K \times 7M}$ in a mini-batch of size $K$

**output** : negative samples in the same mini-batch

1  $SUM^T \leftarrow P^T + N^T$;
2  $P^T \leftarrow P^T / SUM^T, N^T \leftarrow N^T / SUM^T$;
3  Initialize image polarity score $P^I \leftarrow 0, N^I \leftarrow 0$;
4  $EMO \leftarrow Mean(EMO), EMO \in \mathbb{R}^{K \times 7}$;
5  **for** $i \leftarrow 1$ **to** 7 **do**
6     **if** *i is an index of positive emotion categories* **then**
7         $P^I = P^I + EMO_i$
8     **if** *i is an index of negative emotion categories* **then**
9         $N^I = N^I + EMO_i$
10  $P^I = exp(P^I)/(exp(P^I) + exp(N^I))$;
11  $N^I = exp(N^I)/(exp(P^I) + exp(N^I))$;
12  **for** $i \leftarrow 1$ **to** $K$ **do**
13     **for** $j \leftarrow 1$ **to** $KN_S$ **do**
14         $SEN_{T2I}^{ij} = 2 - abs(P_i^T - P_j^I + N_i^T - N_j^I)$
15     **for** $j \leftarrow 1$ **to** $K$ **do**
16         $SEN_{I2T}^{ij} = 2 - abs(P_i^{Ipos} - P_j^T + N_i^{Ipos} - N_j^T)$
17  Sampling negative samples based on probability distribution
    $softmax(SIM + softmax(SEN))$

---

learning as follows:

$$\mathcal{L}_{Intra} = \frac{-1}{N_S} \sum_{i=1}^{N_S} \log \frac{\exp\left(v_{aug1}^i \cdot v_{aug2}^i\right)}{\sum_{j=1}^{N_S} \exp\left(v_{aug1}^i \cdot v_{aug2}^j\right)}. \quad (7)$$

In this manner, we pull apart stickers with similar subject elements in the semantic space, guiding the model to learn discriminative emotion representations.

***2) Inter-Topic Semantic Contrastive Learning.*** Sampling negative stickers solely from the same topic is monotonous. We do not only aim for discriminative sticker representations to be learned but also strive for diversity. To enhance the diversity of negative samples, we introduce inter-topic contrastive learning. In contrast to intra-topic, inter-topic contrastive learning selects candidate stickers under other topics in the same batch as negative samples.

$$\mathcal{L}_{Inter} = \frac{-1}{K \times N_S} \sum_{i=1}^{K \times N_S} \log \frac{\exp\left(v_{aug1}^i \cdot v_{aug2}^i\right)}{\sum_{j=1}^{K \times N_S} \exp\left(v_{aug1}^i \cdot v_{aug2}^j\right)}. \quad (8)$$

The intra-topic and inter-topic contrastive learning together compose the TSCL, which not only guides the model focus on discriminative regions but also ensures diverse negative samples of contrastive learning.

## 3.4  Polarity-based Hard Sample Mining

In this section, we propose a hard sample mining strategy based on the consistency of sentiment polarity. Stickers perform the role of emotion regulation in the dialogue. The sentiment polarity of dialogue and sticker tends to be consistent as verified in Fig. 1. From the process of image-text alignment (*i.e.*, Equ. (3-4)), we can obtain the semantic similarity between image and text modalities:

$$SIM_{I2T} = \hat{\mathbf{V}}_{pos}\hat{\mathbf{W}}, SIM_{T2I} = \hat{\mathbf{W}}\hat{\mathbf{v}}. \quad (9)$$

For each dialogue, we employ an off-the-shelf Chinese text sentiment analysis library, named cnsenti[1], to extract the sentiment polarity score of each utterance. And sum the positive and negative scores for each utterance as the score of the dialogue, denoted as $P^T$ and $N^T$. For each sticker, we extract the polarity score using the image emotion distribution obtained from Section. 3.2 (*i.e.*, $EMO = \mathbf{V}^e E$). Then, we select negative samples based on the semantic similarity scores of sticker and dialogue as well as the sentiment polarity scores. We define a sample to be a hard one if the semantic similarity between the samples is greater and the difference in sentiment polarity is smaller. The above selection strategy is summarized in Algorithm 1.

## 3.5  Training & Inference

In the training phase, our overall objective function is the summation of the above-mentioned formulas:

$$\mathcal{L}_{total} = \mathcal{L}_{I2T} + \mathcal{L}_{T2I} + \mathcal{L}_{SDM} + \alpha \mathcal{L}_{EKD} + \beta \mathcal{L}_{Intra} + \gamma \mathcal{L}_{Inter}, \quad (10)$$

where $\alpha, \beta, \gamma$ are trade-offs between the individual objective functions. We set $\alpha = \beta = \gamma = 0.5$ empirically.

In the inference phase, we discard the Emotion Knowledge Distillation and the Topic-level Semantic Contrastive learning modules, and the model outputs ranked scores for each $s_i$ based on the input $S$ and $U$.

## 4  EXPERIMENTS

### 4.1  Dataset and Metrics

**StickerChat** [12]. The StickerChat dataset collects data from an online chat application. It has $3,516$ sticker topics containing a total of $174,695$ stickers. The 20 utterances before each sticker image are used as the history dialogue to build a dialogue-sticker pair. StickerChat contains $320,168$ pairs for training, $10,000$ pairs for validation, and $10,000$ pairs for testing respectively. The negative samples are constructed by 9 stickers which are randomly sampled from the ground truth sticker topic.

**DSTC10-MOD** [11]. The DSTC10-MOD contains 45000 open-domain conversations with a total of 307 stickers. Following [55], we only adopt the Chinese version of the DSTC10-MOD. Note that since the test set is not publicly available, we use the validation set to evaluate our methods and comparison methods.

**Metrics**. Following the previous works [12, 58], we use the $R_n@K$ ($K = 1, 2, 5$) and Mean Average Precision (*MAP*) to evaluate the proposed method. $R_n@K$ measures whether the ground truth sticker is ranked in the top $K$ of the $n$ candidates. *MAP* measures the mean of the average precision of the total samples. In the following tables, all metrics are shown in percentages, and we omit the % for brevity.

### 4.2  Implementation Details

During testing, there are 10 stickers in each candidate set, of which 9 negative samples are randomly selected from the set containing the

---

[1] https://github.com/hiDaDeng/cnsenti

**Table 1: Comparison of experimental results with previous methods on the StickerChat [12] dataset. Ours denotes the method without the EKD and PHSM modules, Ours* denotes our complete method.**

| Backbone Network | Methods | MAP | $R_{10}@1$ | $R_{10}@2$ | $R_{10}@5$ |
|---|---|---|---|---|---|
| ResNet+Transformer | PSAC [29] | 66.2 | 53.3 | 64.1 | 83.6 |
| | **Ours** | 68.2 | 55.5 | 67.0 | 85.3 |
| | **Ours*** | 72.1 | 60.4 | 71.7 | 88.3 |
| Inceptionv3+Transformer | DAM [58] | 62.0 | 47.4 | 60.1 | 81.3 |
| | SRS [12] | 70.9 | 59.0 | 70.3 | 87.2 |
| | **Ours** | 70.3 | 58.4 | 69.1 | 86.3 |
| | **Ours*** | 72.3 | 60.8 | 71.9 | 86.9 |
| Inceptionv3+GRU | SMN [51] | 52.4 | 35.7 | 48.8 | 73.7 |
| | MRFN [46] | 68.4 | 55.7 | 67.2 | 85.3 |
| | LSTUR [2] | 68.9 | 55.8 | 68.0 | 87.4 |
| | **Ours** | 71.0 | 58.9 | 70.4 | 87.4 |
| | **Ours*** | 72.4 | 60.9 | 72.2 | 87.9 |
| Inceptionv3+LSTM | Synergistic [15] | 59.3 | 43.8 | 56.9 | 79.8 |
| | **Ours** | 69.8 | 57.3 | 68.8 | 87.4 |
| | **Ours*** | 72.5 | 61.4 | 71.6 | 87.4 |
| ViT+BERT | CLIP [41] | 70.9 | 59.1 | 70.3 | 86.8 |
| | ALBEF [25] | 76.8 | 67.0 | 75.6 | 90.0 |
| | **Ours** | 77.6 | 68.4 | 76.6 | 90.3 |
| | **Ours*** | **79.2** | **69.3** | **79.5** | **93.5** |

**Table 2: Comparison of experimental results with previous methods on the DSTC10-MOD [11] dataset.**

| Method | MAP | $R_{10}@1$ | $R_{10}@2$ | $R_{10}@5$ |
|---|---|---|---|---|
| SRS [12] | 50.3 | 30.5 | 54.2 | 71.3 |
| MOD-GPT [11] | 52.3 | 31.2 | 54.8 | 72.1 |
| CLIP [41] | 54.9 | 38.4 | 56.5 | 52.3 |
| MMBERT [55] | 57.7 | 37.1 | 51.3 | 85.2 |
| MMBERT* [55] | 64.3 | 45.9 | 67.0 | 89.5 |
| Ours | **65.0** | **47.2** | **67.1** | **90.1** |

correct sticker. In all of our experiments, we follow the same negative sample construction manner as [12]. When using LSTM [20], GRU [7], and Transformer [47] as text encoder, we use pre-trained word2vec embeddings [26], and the max text length $N_T$ is set to 256. The Transformer approach follows the setting of [12]. For BERT [9], we use the first six layers of pre-trained bert-base-chinese [2], and the max text length $N_T$ is set to 512. For image features, we use the ALBEF [25] pre-trained ViT-B/16 [10] for encoding. For Inceptionv3 [44] and ResNet [18], we use the model parameters provided by timm [50] that is pre-trained on ImageNet [8]. The input image is randomly cropped and then resized to $128 \times 128$. The dimension of the text features $C_T$ and image features $C_V$ is set to 768, and the dimension of common embedding space $C$ is 256. We use the AdamW optimizer with an initial learning rate of $10^{-4}$ and decay to $10^{-5}$ by the cosine schedule. The batch size is set to 4, for a total of $200,000$ training iterations. And validation is performed every $5,000$ iteration. All the experiments are performed on NVIDIA GTX 3090 using PyTorch [38].

## 4.3 Quantitative Analysis

Table. 1 shows the comparison results. We follow the selection of the comparison methods by [12]. For a fair comparison, we divided

[2]https://huggingface.co/bert-base-chinese

**Table 3: Ablation studies on the StickerChat [12] dataset. "Base" indicates ViT+BERT architecture; We set $M = 60$ in EKD module; "Intra" and "Inter" denote the contrastive semantic learning of Intra-Topic and Inter-Topic, respectively.**

| Methods | EKD | Intra | Inter | PHSM | MAP | $R_{10}@1$ |
|---|---|---|---|---|---|---|
| Base | | | | | 76.78 | 67.01 |
| | ✓ | | | | 78.06 | 69.00 |
| | | ✓ | | | 77.20 | 67.71 |
| | | | ✓ | | 77.49 | 67.89 |
| | | ✓ | ✓ | | 77.60 | 68.42 |
| | | | | ✓ | 78.69 | 68.62 |
| | ✓ | | | ✓ | 78.96 | 69.05 |
| Ours | ✓ | ✓ | ✓ | ✓ | **79.23** | **69.38** |

the different methods according to the backbone network of image and text encoders. Meanwhile, in order to analyze the effect of external knowledge (*i.e.*, SER30K and cnsenti), we additionally compared the methods after removing the EKD and PHSM modules (denoted as **Ours**), and the complete methods are denoted as **Our***. Compared with the methods designed for visual question answering (*i.e.*, Synergistic [15] and PSAC [29]), Our method is specifically designed modules for SRS task, and considers emotional association between dialogue and stickers, thus achieving better performance. SMN [51], DAM [58], MRFN [46] and LSTUR [2] are the representative approaches for recommendation task, which are more focused on modeling text as well as semantic information contained in the conversation. However, SRS needs not only to reason about dialogue semantics but also to perceive cross-modal emotion information. That makes them unsatisfactory in terms of sticker selection. SRS [12] is designed for the SRS task. CLIP [41] and ALBEF [25] are generalized pretrained multimodal language models commonly used for retrieval tasks. The previous method achieved best performance of 76.8% on *MAP* and 67.0% on $R_{10}@1$. However, they also ignore emotional information. Our approach utilizes a more concise framework that extracts emotional information from both stickers and sticker-dialogue, resulting in a great performance, *i.e.*, 2.4% in *MAP* and 2.3% in $R_{10}@1$. In addition, our methods achieved superior performance compared to other methods that used the same backbone network, where **Ours*** improved *MAP* by an average of 2.32% and $R_{10}@1$ by 2.86% over **Ours** on the five network combinations. It shows that emotional information is crucial for SRS and our method has excellent generalization capabilities across different networks.

To the best of our knowledge, no previous SRS methods have been validated on both StickerChat and DSTC10-MOD. This may be due to significant differences between the two datasets, such as the number of stickers and the modality of the history dialogue (the history dialogue in DSTC10-MOD contains replyed stickers), which leads to deviations in the basic structure of the model. Therefore, on DSTC10-MOD, we adopt the SOTA method MMBERT [55] as the backbone and compose it with our proposed modules. Table. 2 shows the experimental results of our method on the DSTC10-MOD. Since the stickers in DSTC10-MOD do not have topic annotations, only EKD and PHSM are used in the comparison. And the PBR using only two modules surpasses the previous method. It is worth noting that there are only 307 stickers in the dataset. We suppose the model can learn more diverse emotional information to perform better when the sticker scale increases.

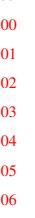
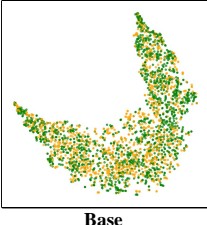
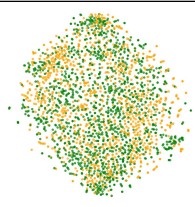
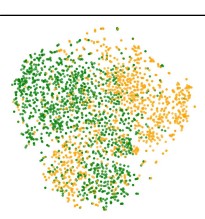

**Figure 4: t-SNE visualization of** $2,000$ **sticker features extracted by image encoder with different methods. black indicates positive polarity samples. Green indicates negative polarity samples.**

**Table 4: Model performance with the different number of clusters** $M$**. The value of** $M$ **is taken every** $10$ **interval from** $10$ **to** $100$**.**

| $M$ | 10 | 20 | 30 | 40 | 50 | Better |
|------|-------|-------|-------|-------|-------|---|
| $MAP$ | 77.60 | 77.64 | 77.82 | 77.62 | 77.82 | |
| $R_{10}@1$ | 68.39 | 68.34 | 68.46 | 68.48 | 68.52 | |
| $M$ | 60 | 70 | 80 | 90 | 100 | |
| $MAP$ | 78.06 | 77.64 | 77.41 | 77.84 | 77.82 | |
| $R_{10}@1$ | 69.00 | 68.30 | 67.96 | 68.73 | 68.54 | Worse |

**Table 5: Performance comparison of different distillation methods. In our method, we take the number of clusters** $M = 60$**.**

| Methods | $MAP$ | $R_{10}@1$ | $R_{10}@2$ | $R_{10}@5$ |
|---------|-------|-----------|-----------|-----------|
| feature distillation | 77.28 | 67.81 | 76.49 | **90.74** |
| logit distillation | 77.41 | 68.25 | 76.41 | 90.36 |
| Ours | **78.06** | **69.00** | **77.36** | 90.63 |

## 4.4 Ablation Studies

In Table. 3, we compare the effect of the proposed modules. The ViT+BERT architecture is used as our "Base" model. It can be seen that the EKD, Intra-Topic, Inter-Topic, and PHSM modules all improve the performance compared to the base model. The TSCL module improved the performance on $MAP$ by 0.82% and on $R_{10}@1$ by 1.41%, and the EKD module achieved a performance increase of 1.28% on $MAP$ and 1.99% on $R_{10}@1$. We believe that the performance improvement of the aforementioned two modules can be attributed to the utilization of more effective sticker representations in our learning process. To verify the hypothesis, in Fig. 4 we visualize the sticker representations using t-Distributed Stochastic Neighbor Embedding (t-SNE).

We can see that the sticker features extracted by the Base architecture are densely clustered and lack discriminative properties. The TSCL strategy allows sticker features to be dispersed in the semantic space, but images with the same sentiment polarity are still blended together. The EKD module allows the encoder to learn emotional knowledge and separate stickers from sentiment polarity. In the first five rows, we use semantic similarity to select negative samples for the matching task, while the sixth row employs the PHSM strategy, which selects hard samples based on the semantic similarity and differences in sentiment polarity of stickers within a batch. In the sixth row, we use the emotion pseudo label extracted by ResNet for

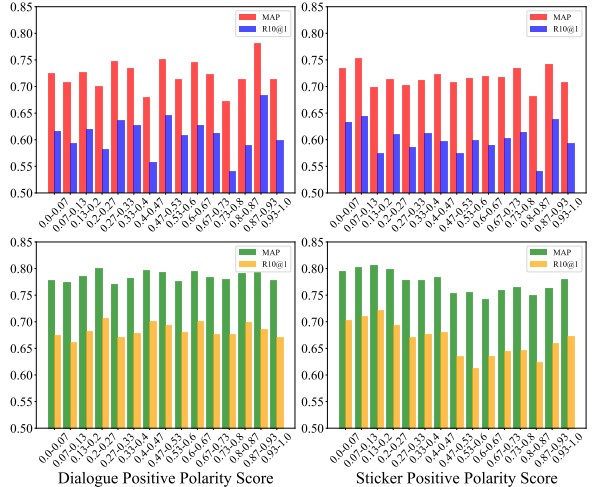

**Figure 5: Left: The effect of dialogue sentiment intensity on model performance. Right: The effect of sticker sentiment intensity on model performance. The upper part shows the trend of SRS [12] and the lower part shows the trend of our approach.**

hard sample mining (*i.e.*, $\mathbf{V}^eE$), and the last two rows use the emotion labels predicted by the model (*i.e.*, $\mathbf{V}W^E$). We finally achieve a model performance of 79.19% in $MAP$ and 69.45% in $R_{10}@1$ using all designed modules, which outperforms the Base model by 2.48% and 2.44%, respectively.

Table. 4 shows the performance of the model with different cluster numbers $M$. The difference in $MAP$ between the highest point $M = 60$ and the lowest point $M = 80$ is 0.65% and the difference in $R_{10}@1$ is 1.04%, which indicates the robustness of the model at different $M$. Furthermore, we found small variations in model performance at $M <= 50$ and larger fluctuations after $M > 60$. We conclude that this is due to the small sample size of some emotion categories in the SER30K dataset (*e.g.*, 211 disgust, 826 fear, *etc.*). As the number of clusters rises, the clustering effect of these categories decreases, which in turn affects the cluster centroids.

Table. 5 shows the performance comparison of different distillation methods. The table shows that feature distillation has the worst results, We conclude that since emotion is high-level semantic information, it cannot be fully understood using feature distillation. The logit distillation performed better on $MAP$ and $R_{10}@1$ metrics. And our EKD module achieved the best performance, outperforming logit distillation by 0.65% and 0.75% on $MAP$ and $R_{10}@1$, respectively. The number of soft target dimensions generated for each sample by logit distillation is equal to the number of emotion categories. Due to the topic structure of stickers, multiple characters will exist under the same emotional category. Therefore we use the ClusterFit approach to generate more fine-grained soft targets whose dimensionality is not restricted to emotional categories. It makes the sticker representations learned by our EKD module more diverse.

## 4.5 Effect of Sentiment Intensity

Since our model can perceive the emotion of the stickers, the emotion intensity of the sample should influence the performance of the

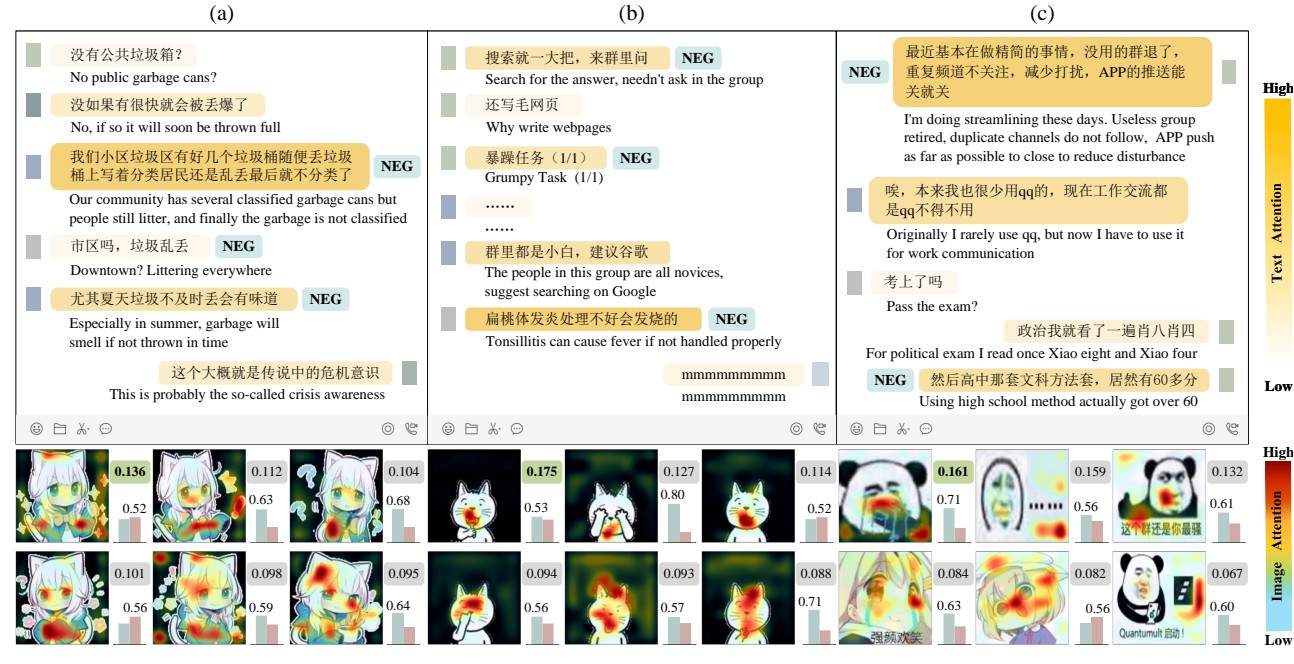

**Figure 6: The visualization results on the validation set. For each sample, we show the top-6 stickers predicted by our model. The background color of the utterances indicates how much attention they receive. The number on the right of stickers represents the ranking score (ground truth labeled with green background), and the bar represents the sentiment score (blue for negative, red for positive).**

model. To verify this hypothesis, Fig. 5 illustrates the performance variation of the comparative method SRS [12] and our method as the sample sentiment intensity changes. We use continuous scores between 0 and 1 to discript sentiment intensity, where 0 represents negative and 1 represents positive. The left part shows the impact of dialogue sentiment intensity on model performance. SRS performance exhibits irregular fluctuations, whereas the accuracy of ours remains relatively stable. The right part shows the impact of sticker sentiment intensity on model performance. SRS still lacks a regular trend, while ours demonstrates a noticeable V-shaped pattern. When the sentiment intensity is intense, the $MAP$ and $R_{10}@1$ of the model reach a maximum of 80.58% and 72.05%, respectively. But when the sentiment intensity is dim (*i.e.*, in the middle interval), the $MAP$ and $R10@1$ of the model reach a minimum of 74.15% and 61.15%. This indicates that our model pays more attention to the emotion of stickers, and it is more likely to make the right selection when the sentiment intensity of stickers is strong. Furthermore, the lack of a clear relationship between the performance variation of SRS and sentiment intensity suggests that the previous methods do not effectively capture emotional information. Emotions need to be explicitly modeled to achieve optimal utilization.

## 4.6 Qualitative Results

Fig. 6 shows the results of the visualization on the validation set. In the dialogue, sentences that contain sentiment tend to receive higher attention scores from the model. In stickers, regions that are given more attention by our model are typically critical for expressing emotions. For example, in the ground truth stickers, our model focuses on

the gesture and surrounding symbols in (a), while in both (b) and (c), facial expressions are the main focus. In addition, there is a strong consistency between dialogue sentiment and sticker sentiment. The stickers with higher scores mostly have the same sentiment as the dialogue. It indicates that our model not only pays close attention to the sentiment conveyed in dialogues and stickers but also considers the cross-modal sentiment association. It was observed that some image sentiments are predicted incorrectly, which we attribute to the insufficient quality of the pseudo-labels generated by the teacher model. A better teacher model may be able to improve the model's performance.

## 5 CONCLUSION

In this paper, we explore the utilization of emotional information on the SRS task. Intuitively, we need to perceive the emotion of the dialogue and sticker before sending the appropriate sticker. It is challenging to perceive and utilize unimodal and cross-modal emotional information. We consider both emotional and semantic information and design an Emotion Knowledge Distillation module as well as a Topic-level Semantic Contrastive Learning module to obtain discriminative and diverse sticker features. Besides, we improve the hard negative sampling method through the Polarity-based Hard Sample Mining module, which constructs hard negatives based on semantic similarity and sentiment polarity differences during the training phase. Extensive experiments prove the validity of each designed component. Our work provides an emotional perspective on the SRS task, and we also expect to facilitate future research on sticker recommendations.

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
