# OpenReview forum: "Perceive before Respond: Improving Sticker Response Selection by Emotion Distillation and Hard Mining"
_acmmm.org/ACMMM/2024/Conference — MM2024 Poster_

### Official Review · Reviewer_kmPd · 2024-05-06

**Rating:** 4
**Confidence:** 2

**Summary:**

The paper introduces a novel approach, "Perceive before Respond" (PBR), for improving Sticker Response Selection (SRS) in online chatting. It addresses the challenge of predicting the most relevant sticker image in response to a dialogue by leveraging emotional information from stickers and dialogues. The proposed approach includes an Emotion Knowledge Distillation (EKD) module, a Topic-level Semantic Contrastive Learning (TSCL) module, and a Polarity-based Hard Sample Mining (PHSM) module. These modules enhance emotion comprehension, diversity, and discriminative features of stickers, leading to improved sticker response selection.

**Strengths:**

1. The paper introduces a novel approach, PBR, which considers emotional information from stickers and dialogues, enhancing the understanding and selection of stickers in online chatting.
2. The EKD module effectively distills emotion knowledge from a large-scale sticker emotion recognition dataset, improving emotion comprehension in the model.
3. The TSCL module improves the diversity and discriminative features of stickers by considering intra-topic and inter-topic contrastive learning.
4. The PHSM module introduces a novel strategy for hard sample mining based on sentiment polarity, improving the quality of negative samples for the model.

**Limitations:**

1. The paper mentions that existing SRS datasets lack emotion annotations, which may limit the generalizability of the proposed approach. It would be beneficial to explore how the approach performs on datasets with varied emotional annotations.
2. The paper provides extensive experiments to verify the effectiveness of each proposed component. However, it would be helpful to include a more in-depth analysis of the limitations of the approach, such as failure cases or scenarios where the approach may not perform as well.
3. While the approach aims to improve sticker response selection based on emotional relevance, it may be challenging to interpret or explain why certain stickers are chosen over others. User preferences for stickers can be subjective and may not always align with the model's predictions, which could impact user satisfaction and engagement in online conversations.
4. The reliance on the SER30K dataset for emotion knowledge may limit the generalizability of the approach to other sticker datasets or domains.
5. The paper focuses on the SRS task, and it's unclear how well the proposed method would perform on other related tasks or in real-world scenarios with varying sticker usage.

**Suitability:**

3

---

### Official Review · Reviewer_1uwf · 2024-05-23

**Rating:** 2
**Confidence:** 3

**Summary:**

The paper presents a "Perceive before Respond" (PBR) paradigm for Sticker Response Selection (SRS), enhancing emotion understanding through knowledge distillation and improving sticker representation via contrastive learning and cross-modal sentiment association. However, there are some significant issues.

**Strengths:**

1. The authors explore the utilization of emotional information on the SRS task.

2. The motivation and methods are easy to understand.

3. Experimental results demonstrate its robust localization and question-answering performance.

**Limitations:**

1. The core contribution of the paper lies in addressing the three challenges in the sticker selection task, the novelty is limited, in fact, some work has already solved these existing challenges. For example, [1] has already pointed out that stickers vary greatly in style, leading models to learn robust representations for the stickers following various distributions.

2. The author claims in the second challenge that the visual representations extracted from stickers under the same topic are very similar, which would lead to the model's incorrect predictions. I think this idea is intuitive and has not been well verified. The existing MLLMs would not mistakenly judge a smiling face and a crying yellow one in Figure 2 (b). Moreover, the same sticker can also convey different emotions. For example, crying stickers can express sadness in one context and be used to convey being touched in another.

3. Currently, the efficacy of generation models is widely acknowledged. Does the author consider generating stickers directly with their proposed methods, like [2] [3].

4. The novelty of the method is limited and not convincing; it merely involves a straightforward combination of existing modules. The ablation study results show that without the EKD and PHSM modules, which the authors highlighted in the Introduction as addressing key challenges, there is no significant improvement in the experimental outcomes (i.e., the results of R$_10$@1 only improve from 68.42 to 69.38, as shown in Table 3). This does not substantiate the effectiveness of the method or the validity of the challenges presented.

5. Upon a closer look at the comparative methods in the experiments, they are all too outdated. MMBERT merely utilizes Bert for this task in a straightforward manner, and ALBEF is a backbone model the authors use. Such a comparison is quite unfair. Methods based on the latest Multimodal Large Language Models (MLLM) should be considered for comparison.

6. What does the MMBERT* mean in Table 2? The authors should describe this in the caption of the table.

[1] Ge F, Li W, Ren H, et al. Towards Exploiting Sticker for Multimodal Sentiment Analysis in Social Media: A New Dataset and Baseline[C]//Proceedings of the 29th International Conference on Computational Linguistics. 2022: 6795-6804.

[2] Zhang Y, Kong F, Wang P, et al. StickerConv: Generating Multimodal Empathetic Responses from Scratch. ACL 2024.

[3] Wang H, Lee R K W. MemeCraft: Contextual and Stance-Driven Multimodal Meme Generation[C]//Proceedings of the ACM on Web Conference 2024. 2024: 4642-4652.

**Suitability:**

2

---

### Official Review · Reviewer_8Byz · 2024-05-26

**Rating:** 4
**Confidence:** 3

**Summary:**

This paper proposes a "Perceive before respond" training paradigm to address the challenge of sticker response selection. This paradigm has three main contributions:
1. The paper introduces an emotion knowledge distillation module to extract sticker emotion knowledge from the SER30K sticker emotion recognition dataset, using the K-Means method to address the issue of single features inadequately describing emotion categories.
2. The authors propose topic-level emotion contrastive learning, including both intra-topic and inter-topic emotion contrastive learning.
3. The paper introduces a polarity-based hard sample mining method to improve the quality of negative sampling.

**Strengths:**

1. This paper proposes a novel training framework that effectively assists models in utilizing both unimodal and cross-modal emotional information.
2. A hard sample mining method based on sentiment polarity is introduced to improve the quality of negative sampling.
3. The authors conducted comprehensive experimental evaluations and ablation analyses to demonstrate the effectiveness of the framework and the efficacy of individual components.

**Limitations:**

The methods used in the emotion knowledge distillation module and the contrastive learning module within the framework lack sufficient innovation

**Suitability:**

3

---

### Meta-Review · Area_Chair_qD4v · 2024-06-30

**Recommendation:** Accept (Poster)
**Confidence:** 5

**Metareview:**

This paper proposes a novel "Perceive before respond" (PBR) training paradigm to address sticker response selection (SRS), introducing an emotion knowledge distillation (EKD) module, topic-level emotion contrastive learning (TSCL), and a polarity-based hard sample mining (PHSM) method. The approach effectively leverages unimodal and cross-modal emotional information, with comprehensive experimental evaluations demonstrating its robust performance. However, the innovation of the EKD and contrastive learning modules appears limited, as similar challenges have been addressed in prior work. Additionally, the efficacy of visual representations and emotional categorization under different contexts remains inadequately validated. The reported improvements from EKD and PHSM modules are marginal, raising questions about their practical impact. Comparative experiments utilize outdated methods, and more recent multimodal large language models (MLLMs) should be included for a fair assessment. To address these concerns in the camera-ready version, it is recommended that a more detailed innovation analysis be provided, failure case studies should be included, and user subjectivity in sticker preferences should be considered.